# Impact of Preventive Strategies on HPV-Related Diseases: Ten-Year Data from the Italian Hospital Admission Registry

**DOI:** 10.3390/cancers15051452

**Published:** 2023-02-24

**Authors:** Vincenzo Restivo, Giuseppa Minutolo, Marianna Maranto, Antonio Maiorana, Francesco Vitale, Alessandra Casuccio, Emanuele Amodio

**Affiliations:** 1Department of Health Promotion, Maternal and Infant Care, Internal Medicine and Medical Specialties (PROMISE) “G. D’Alessandro”, University of Palermo, 90127 Palermo, Italy; 2HCU Obstetrics and Gynecology, ARNAS Ospedale Civico Di Cristina-Benfratelli Hospital, 90127 Palermo, Italy

**Keywords:** HPV-related disease, cervical cancer, vaccination, screening, effectiveness, real world data, hospital admission, communication, healthcare workers, Italy

## Abstract

**Simple Summary:**

HPV-related diseases are mainly represented by cancers. Furthermore, real world data with respect to the effects of primary and secondary preventive strategies are lacking. Therefore, the aim of this study is to assess the effectiveness of preventive strategies in accordance with Italian HPV-related hospital admissions. From 2008 to 2018, there was a decrease (APC = −3.8%) in all HPV-related diseases. The increase in cervical cancer screening adherence was related to a decrease in invasive cervical cancer and an increase in HPV vaccine coverage, which was found to arise from a decrease in “in situ” cervical cancer. In this study, the need to improve the acceptance of preventive strategies for HPV-related diseases, as well as the homogenous information furnished by all healthcare workers involved in their promotion (e.g., gynecologists, general practitioners, pediatricians) is highlighted.

**Abstract:**

Human papillomavirus (HPV)-related diseases are still a challenge for public health. Some studies have shown the effects of preventive strategies on them, but studies at the national level are few in number. Therefore, a descriptive study through hospital discharge records (HDRs) was conducted in Italy between 2008 and 2018. Overall, 670,367 hospitalizations due to HPV-related diseases occurred among Italian subjects. In addition, a significant decrease in hospitalization rates for cervical cancer (average annual percentage change (AAPC) = −3.8%, 95% CI = −4.2, −3.5); vulval and vaginal cancer (AAPC = −1.4%, 95% CI = −2.2, −0.6); oropharyngeal cancer; and genital warts (AAPC = −4.0%, 95% CI = −4.5, −3.5) was observed during the study period. Furthermore, strong inverse correlations were found between screening adherence and invasive cervical cancer (r = −0.9, *p* < 0.001), as well as between HPV vaccination coverage and in situ cervical cancer (r = −0.8, *p* = 0.005). These results indicate the positive impact of HPV vaccination coverage and cervical cancer screening on hospitalizations due to cervical cancer. Indeed, HPV vaccination also resulted in a positive impact on the decrease in hospitalization rates due to other HPV-related diseases.

## 1. Introduction

Human papillomavirus (HPV) is still the main infection that occurs via sexual activities [1,2,3,4,5,6]. HPV is also the etiological factor that is the most relevant for cervical cancer [2,3,7,8]. Worldwide, the number of HPV genotypes that have, until now, been identified is almost 200 [1,2,4,9,10]. These are classified into low-grade (the most widespread HPV 6 and 11) and high-grade (such as HPV 16 and 18) risk with respect to malignant oncological diseases [10,11]. Due to the high clearance capacity of the human immune system [3], only 10% of HPV infections progress to precancerous and cancerous lesions [8,12]. In addition, certain age classes, determined according to sex, were found to be more affected than others: most female patients were between 35–44 years old, whereas the males were mostly between 55–74 years old [13].

Cervical cancer affects nearly 7–35/100,000 females worldwide per year [3]. In addition, the mortality rates change in accordance with socio-economic conditions [8]. Moreover, several instances of scientific data have shown that nearly 100% of cervical cancer cases were HPV-positive [2,8].

Currently, when estimating Italian data, the HPV-related disease burden has remained a challenge [6]. Furthermore, to date, only a few studies at the national level are available. The latest update in Italy regarding hospitalization due to HPV-related diseases, at first diagnosis, included data from 2001 to 2012, in which 24.0% of Italian cases were in relation to HPV-related diseases that needed hospitalization [13]. In detail, the hospitalization rate in Italy due to cervical cancer when first diagnosed, was 15.6/100,000 females per year during the 2001–2012 period, whereas the in situ cervical cancer rate was 17.6/100,000 females per year [13]. Moreover, it should be highlighted that most of the epidemiological data were obtained at the local level [6]. During the 2008–2011 period, two Italian regions, Veneto and Marche, reported similar hospitalization rates for HPV-related diseases (49.4/100,000 vs. 48.4/100,000), with a decrease of 30.0% and 15.0%, respectively [6].

Other HPV-related oncological diseases had a variable trend throughout Italy with regards to both genders. Considering oropharyngeal cancer, the Italian trend regarding the hospitalization rate was nearly four times higher among men (16.0/100.000 vs. 3.9/100,000) [13]. Penile cancer had the lowest hospitalization rate, with a value of 1.8/100,000 men per year in Italy [13]. Genital warts (GW) were another HPV-related infectious disease, whose onset occurs after 3–35 weeks from HPV 6 and 11 infections in 90% of teenagers and adults. GW are more frequently observed in low-income countries [14], even if the different reporting methodology of clinical data makes it difficult to compare either morbidity or mortality rates [14,15,16]. In Italy, the annual hospitalization rate was 7.5–8.5/100,000 in the years 2001–2012 [13].

The most efficient prevention strategies for cervical cancer are HPV vaccination and screening [4,8,17,18]. Since 2009, all 12-year-old Italian females were the target population for the HPV mass vaccination program (MVP) [19]. Furthermore, males can also receive the vaccine within the MVP since the 2005 birth cohort. Currently, the MVP makes available the HPV 9 valent vaccine (instead of the HPV 4 vaccine), which covered nearly all the most widespread HPV genotypes involved in cancer [3,6]. On the other hand, national organized screening for cervical cancer started in 1996 in a few Italian centers, while the screening was voluntary-based beforehand [20]. Currently, the screening is offered to all Italian females from 25 years of age up to 64 years of age. Among Italian regions, Northern Italy had the best screening uptake [21], increasing the early diagnosis of cervical cancer and consequently lowering the mortality rates [7,21].

According to the World Health Organization, cervical cancer could be eliminated by 2030, if nearly 90% of female adolescents are vaccinated against HPV, if 70% of females aged between 35 and 45 years old perform cervical cancer screening, and finally if 90% of female patients receive healthcare assistance for either pre-cancerous lesions or cervical cancer at any stage. It is expected that reaching these goals will reduce new cases of cervical cancer to lower than 4/100,000 [22]. In order to detect the impact of these preventive measures, it should be important to analyze the hospitalization trend of cervical cancer and other HPV-related diseases. A similar approach was adopted in Sicily where cervical cancer had the highest hospitalization rate in 2007–2017 (106.3/100,000). In this setting, prevention strategies, such as cervical screening and HPV vaccination, progressively reduced the impact on hospital admission regarding cervical cancer [7].

The main aim of this study is to assess hospitalization rates for HPV-related diseases in Italy. The second aim is to focus on the impact of prevention measures on hospitalization rate reductions for cervical cancer in the Italian setting.

## 2. Materials and Methods

### 2.1. Study Design

In this descriptive study, the hospitalization trends due to HPV-related diseases in Italy between 2008 and 2018, excluding the administrative region of Sicily, were evaluated. The source of the data was the Italian hospital discharge records (HDRs) database, which is a medical recording system for each admission to Italian hospitals since 1991 [23]. The main objective of HDR is for healthcare performance payments, followed by disease classification in fixed fee categories that are named diagnoses related groups (DRGs), in which homogeneous diagnoses were included [24,25,26]. However, after several implementations regarding disease classification and the clinical information of discharged patients [23,25,27,28,29,30,31,32,33], the variables contained in HDRs can be used both for epidemiological and healthcare planning purposes. The variables contained in the HDRs are socio-demographic (such as sex, date of birth, birthplace, educational level, citizen, residence, marital status, etc.) and medical (healthcare setting; admission and discharge dates; diagnoses and interventions encoded following the International Classification of Disease 9th edition; Clinical Modification [ICD9-CM]; and outcome of discharge) [23,25,31,32,34].

### 2.2. Target Population

According to the Italian Statistics Institutes, an annual average of 54,992,245 inhabitants have lived in Italy (excluding Sicily) in the study period with a percentage of 51.5% females [35]. Among them, it has been estimated that 79.6% (from a total of 15,281,090) of the female target population, aged 25–64 years old, had a cervical cancer screening from 2008 to 2017 [36,37,38]. When focusing on the female birth cohorts of 1997–2006, the HPV vaccination mean coverage retrieved by the Italian Health Ministry during the period 2009–2018 was 67.2% (from a total of 1,374,993) [19,39,40,41,42,43,44,45,46,47]. On the other hand, vaccination coverage against HPV in the male birth cohorts of 2003–2006 were either too low, or fluctuated every year between 2016 and 2018. The highest percentage of vaccination adherence was reached in 2018, with a value of 17.5% in Italian males [45,46,47].

### 2.3. Inclusion Criteria

The identification of HPV-related hospital admissions was in accordance with the following codes of ICD9-CM, as reported in other studies [7,34]: condyloma acuminatum (078.11), head and neck cancers (140.0–149.9, 195.0, 230.0, and 235.1), anal cancers (154.2–154.8, 230.5–230.6), cervical cancers (180.0–180.9, 233.1, 622.1, 654.6, and 795.0–795.1), genitourinary tract cancers—vagina, labia, clitoris (184.0–184.8)—and penile cancers (187.1–187.9, 233.5).

In order to increase the accuracy of cervical cancer identification, as per the inclusion criteria, the following interventions codes were also considered: cervix conization (67.2), cervical lesion cauterization (67.32), and cervical lesion cryosurgery [7].

Both primary and secondary diagnoses or the intervention codes of the HDRs were considered. The quality of HDRs was checked in order to remove those with missing, duplicated, or inaccurate data.

### 2.4. Statistical Analysis

Absolute and relative frequencies were calculated for qualitative variables (sex, marital status, citizenship, administrative Italian region of residence, and HPV-related diseases). The skewness and kurtosis tests were used to analyze the continuous variables (i.e., the age and length of hospital stay in days) in order to choose the mean and standard deviation (SD) for normal distribution. Otherwise, the median and interquartile range (IQR) were used. A nonparametric K-sample test on the equality of medians was performed for the annual median of the patients’ age, as well as the median in relation to the length of stay, in days, per year. Annual and total hospitalization rates for each HPV-related disease (cervical cancer; vulval and vaginal cancer; penile cancer; oropharyngeal cancer; anal cancer; and genital warts) were calculated per 100,000 inhabitants using the related attributable fractions. They were previously reported in another manuscript: cervical cancer 100%; genital warts 100%; anal cancer 88%; vulval and vaginal cancer 77%; penile cancer 50%; oropharyngeal cancer 26% [7]. The Breusch–Pagan/Cook–Weisberg tests were utilized for the heteroskedasticity evaluated variances of annual hospitalization rates.

The national percentages of cervical cancer screening were obtained from the PASSI data [36,37,38,39]. In addition, HPV vaccination coverage by the Italian Health Ministry [19,39,40,41,42,43,44,45,46,47] was calculated. Pearson’s correlation (r) and determination (r2) coefficients were used to measure the correlation of the hospitalization rate due to invasive and in situ cervical cancer. This was conducted with the percentage of cervical cancer screening adherence and HPV vaccination coverages, respectively [48,49,50].

The annual percent change (APC) and the average APC (AAPC) were evaluated by the Joinpoint Regression Program, Version 4.9.1.0. (April, 2022; Statistical Research and Applications Branch, National Cancer Institute), following the methodology described by Kim et al. [51]. In the Joinpoint Regression Program, a constant variance was chosen in the case of homoscedasticity among the annual hospitalization rates, the adherence to cervical cancer screening, and the HPV coverages. Otherwise, a calculation of standard error (SE) was performed and included before the Joinpoint analysis [51]. The best model was chosen in accordance with the permutation test. Confidence intervals at 95% (95% IC) and *p*-values were reported. The *p*-value was considered statistically significant when α ≤ 0.05. Stata/SE 14.2 (Copyright 1985–2015, StataCorp LLC, 4905 Lakeway Drive, College Station, TX 77845, USA. Revision 29 January 2018) was used in order to perform statistical analysis.

## 3. Results

Overall, the HDRs for HPV diseases were 670,367 during the study period, while those attributable to HPV infection, according to attributable fractions, were 483,373 (72.1%). The HDRs that reported death during the hospitalization were 2.2% (*n* = 15,794).

Table 1 showed the socio-demographic and clinical characteristics of hospitalized patients. The median age was 54 years old (IQR = 41–68). The most representative demographic were females (66.6%, *n* = 446,435) and married subjects (42.0%, *n* = 300,235). Many HDRs reported Italian citizenship (92.2%, *n* = 617,826), whereas refugees and resident foreigners in Italy were 1.0% (*n* = 6635) and 6.8% (*n* = 45,906), respectively.

HDRs, due to cervical cancer, had the highest percentage of admissions (43.5%, *n* = 291,368). Conversely, instances of penile cancer had the lowest percentage (2.4%, *n* = 15,804). The median length of a hospital stay was 5 days (IQR = 3–11).

Table 2 details the hospitalization rates due to HPV-related diseases, per year, between 2008 and 2018. In addition, the trends of these declined for all of them. Among the cancers, the highest reduction was found for cervical cancer with a value of 31.2% (from 115.1 to 79.2 per 100.000). This was followed by oropharynx cancer with a reduction of 30.9% (from 12.3 to 8.5 per 100.000), and then anus cancer with a reduction of 17% (from 7.1 to 5.9 per 100.000). Overall, the condition with the highest reduction was GW with a value of 33.4% (from 11.7 to 7.8 per 100.000).

As reported in Figure 1, cervical cancer had a significant decline in hospitalization rates (AAPC = −3.8%, 95% CI = −4.2, −3.5; *p* < 0.001), with a Joinpoint in 2011 and a higher reduction in the period 2008–2011 (AAPC = −5.8%,95% CI = −7.1, −4.5; *p* < 0.001) than in comparison to 2011–2018 (APC = −3.0%, 95% CI = −3.3, −2.6; *p* < 0.001). Another Joinpoint was shown for penile cancer, which increased in the period 2008–2013 (APC = 0.5% 95% CI = −0.7, 1.7; *p* = 0.352) and then decreased in the period 2013–2018 (APC = −1.4%, 95% CI = −2.6, −0.3; *p* = 0.023). Furthermore, anal cancer highlighted a Joinpoint with a decreasing trend from 2008 to 2014 (APC = −3.7%, 95% CI = −4.7, −2.8; *p* < 0.001) and a slightly increasing trend in the period 2014–2018 (APC = 1.3% 95% CI = −0.6, 3.3; *p* = 0.136). The other diseases saw a decreasing trend without Joinpoint. Additionally, the highest value in AAPC was for genital warts (AAPC = −4.02%), followed by oropharyngeal cancer (AAPC = −3.99%).

There was a statistically significant increase in the median age of patients hospitalized for cervical cancer from 44 years in 2008 to 46 years in 2018 (*p* < 0.001). Similarly, the length of hospital stays increased significantly from 2008 (3 [IQR = 2–7]) to 2018 (4 [IQR = 2–7]) (*p* < 0.001; Table 3). Instances of female death due to cervical cancer during hospitalization were 2085 (0.72%); their median age was 65 years old (IQR = 52–77) (data not reported in the table).

The hospitalization rate for invasive cervical cancer (Figure 2) was 30.9 per 100,000 female inhabitants in the whole study period. All ages had a significant decline in hospital admission rates (AAPC = −3.5%, 95% CI = −3.1, −24.3). The highest decline in invasive hospitalization was for females over 75 years old, with an AAPC of −6.1%, (95% CI = −7.7, −4.6). This was followed by females aged 65–74 years old (AAPC = −4.8%, 95% CI = −6.0, −3.7) and females aged 45–54 years old (AAPC = −4.4%, 95% CI = −5.2, −3.6). On the other hand, there was an increase in those aged 25–34 years old (AAPC = 0.4%, 95% CI = −1.3, 2.1). 

When considering in situ cervical cancer (Figure 3), the hospitalization rates of females aged 15–24 years old had the highest significant decrease with an AAPC = −8.9% (95% CI = −11.7, −6.0) between 2008 and 2018. The age class with the next highest decrease were those over 75 years old (AAPC = −6.7%, 95% CI = −9.8, −3.6), followed by those aged 65–74 years (AAPC = −5.4%, 95% CI = −7.3, −3.4). On the other hand, there was an increase in hospital admissions in the age class of 55–64 years (AAPC = 2.0%, 95% CI = 0.3–3.8).

During the period 2008–2017, cervical screening uptake among females aged 25–64 years old was 79.6%, with significant increases observed during the period (AAPC = 0.5%, 95% CI = 0.2, 0.7, *p* < 0.001). This was strongly inversely correlated with hospitalization rates due to invasive cervical cancer (r = −0.9, *p* < 0.001), as shown in Figure 4.

HPV vaccination between 2009 and 2018 had a coverage of 67.2% with significant increases observed during the period (AAPC = −2.0%, 95% CI = 1.1, 2.9; *p* = 0.001). This trend was strongly inversely correlated with hospitalization rates due to in situ cervical cancer (r = −0.8, *p* = 0.005), as shown in Figure 5.

## 4. Discussion

This study showed the hospitalization trends of HPV-related diseases from 2008 to 2018 in Italy. Despite its significant decrease along the whole period with an AAPC = −3.8 (95% CI = −4.2, −3.5; *p* < 0.001), cervical cancer was the HPV-related disease with the highest hospitalization rate (93.5/100,000). The Italian AAPC for cervical cancer was milder in comparison to the Sicilian data, in which the hospitalization rate for cervical cancer had a statistically significant AAPC = −9.9%, [7]. Furthermore, the Italian reduction in HPV cervical cancer from 2008 to 2018 was slightly higher than that observed in the period 2001–2012 (APC = −2.9%, 95% CI = −3.8, −2.1) [13]. This can be understood as a further demonstration of the impact of preventive strategies on HDRs.

The females’ median age and length of hospital stay slightly increased during the study period (*p* < 0.001). This was similar to the Sicilian data and may be an indirect effect of the preventive strategies. Indeed, vaccine availability was available to people born after 1996, and the older females did not have the benefit of HPV vaccination. In addition, these older females could have received the diagnosis of cervical cancer later, when it began to be symptomatic, and could have had other diseases or complications, thus resulting in extending the length of their hospital stay [7].

The rate for cervical cancer was similar to that reported by a Swedish study, which was based on registry data among females who were not vaccinated against HPV (93.5/100,000 vs. 94/100,000) [52]. Another study in England has recently shown that the incidence rate ratio for cervical cancer was inversely correlated to the HPV vaccination recipients’ age, which was lower among those vaccinated at the age of 8 years old than in other age cohorts (0.13 vs. 0.38 in the 10–11 years old cohort and 0.66 in 12–13 years old cohort) [49].

A globally observed reduction in hospitalization rates due to in situ and invasive cervical cancer was detected in most of the age classes, even if the rate was higher among those older than 75 years old for invasive cancer, and those aged 15–24 for in situ cancer. In addition, the significant increase in HPV vaccination coverage among females aged 12–21 years old and adherence to cervical cancer screening among females aged 25–64 years old seem to reduce the hospitalization rates due to in situ and invasive cervical cancer, respectively. This could suggest an impact of HPV vaccination and cervical cancer screening on cervical cancer reduction [7,8,17]. Furthermore, it could be related to the availability of preventive healthcare services for either HPV vaccination centers, or for cervical cancer screening throughout Italian regions. As well as the spatial-temporal variability of the HPV vaccination coverage [19,39,40,41,42,43,44,45,46,47], most of the Southern Italian regions had access to the most updated, high valent HPV vaccines before the Northern regions [19]. Additionally, the cervix screening uptake had variable percentages from one Italian region to another, and this could influence the cervical cancer trends [36,37,38]. On the other hand, lifestyles, individual risks, or protective factors could also affect the onset of cervical cancer, such as smoking, sexual promiscuity, or genetic polymorphisms [53].

According to the newest meta-analysis data, whatever vaccine valence was undertaken against HPV, nearly 100% of cervical lesions and infections due to some HPV serotypes (such as 16 and 18) can be avoided, thus showing an almost decennial antibody protection in several trial studies [54]. This supports the results of this study, whereby the long-lasting effectiveness of more than 10 years of vaccine introduction in Italy is confirmed once again [55]. For these reasons, HPV vaccination should be promoted by all healthcare workers and should reach all eligible HPV vaccine recipients [56,57]. This will improve coverage in the at-risk population, thereby decreasing the burden of HPV-related malignancies such as cervical cancer.

Hospitalization rates due to oropharyngeal cancer in Italy decreased constantly during the period 2008–2018 (APC = −4.0%, *p* < 0.001). Nevertheless, these low trends may be attributed to HPV vaccination [58]. Indeed, sexual practices can transmit the virus, especially HPV16, and its survival is possible for the anatomy of the oropharynx district. Therefore, the link to cancer in this area cannot be excluded [5]. On the other hand, oropharyngeal cancer could be an example of an oncological disease that is caused by either HPV or damaging substances (such as alcoholic drinking and smoking) [5,58]. Therefore, other studies need to be conducted, where the impact of alcoholic drinks and smoking habits among people who are either vaccinated or unvaccinated against HPV, should be evaluated.

Penile cancer had the lowest hospitalization rate in Italy during the years 2008–2018 because of its rarity, as is shown in the European data [59]. However, the penile cancer rate was higher than what was reported in European and Northern American countries (2.7/100,000 vs. 1.0/100,000) [59]. Despite its severity, there is still no prevention strategy for this oncological disease and HPV-vaccination may only have an effectiveness of 33.3% [59]. The low effectiveness can be due to the fact that there are other risk factors for penile cancer, which can be either genetic, inflammatory, or environmental [59]. Moreover, the HPV-vaccination coverage in Italy for adolescent males is too low and too recent in order to correlate to penile cancer trends or to consider as an effective prevention measure.

This study had some limitations. HDRs could underestimate the real trend of these HPV-related diseases because some of them, especially the less severe ones such as cervical cancer, may have been treated in an outpatient setting, whose clinical and administrative sources of data were not available and not included in this study. In detail, precancerous cervical lesions are more frequently treated in outpatient settings during colposcopy, which is an outpatient procedure, according to national laws [60,61,62]. Furthermore, the inappropriate use of codes for HPV-related diseases could not be excluded. Moreover, the HPV attributable fraction of HDRs is not equivalent to an exact measure of HPV status. Although the previously reported limits are different, this study suggests that hospital discharge records can represent a very efficient way for evaluating the trends of HPV-related diseases, as well as the direct/indirect benefits of primary (HPV vaccination) and secondary (cervical cancer screening) prevention.

## 5. Conclusions

Cervical cancer is the HPV-related disease with the highest impact on women’s health. HPV vaccination and cervical cancer screening in Italy are associated with a reduction in hospitalization rates due to in situ and invasive cervical cancer. Additionally, other HPV-related diseases reported in this study decreased globally. Further studies should confirm our findings investigating, at an individual level, the effectiveness of preventive strategies. While waiting for these results, our analyses provide evidence-based data to support adherence to HPV preventive strategies and thus achieve the elimination of HPV.

## Figures and Tables

**Figure 1 cancers-15-01452-f001:**
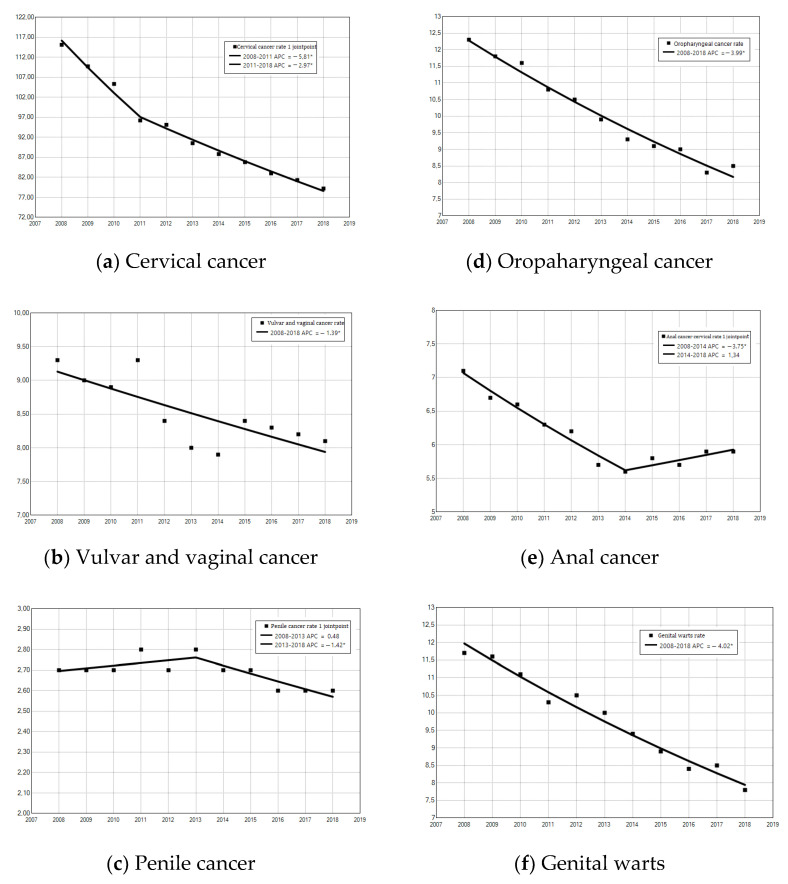
Annual percentage change (APC) of hospitalization due to HPV-related diseases: (**a**) cervical cancer, (**b**) vulval and vaginal cancer, (**c**) penile cancer, (**d**) oropharyngeal cancer, (**e**) anal cancer, and (**f**) genital warts. * APC significantly different from 0 at the α = 0.05 level.

**Figure 2 cancers-15-01452-f002:**
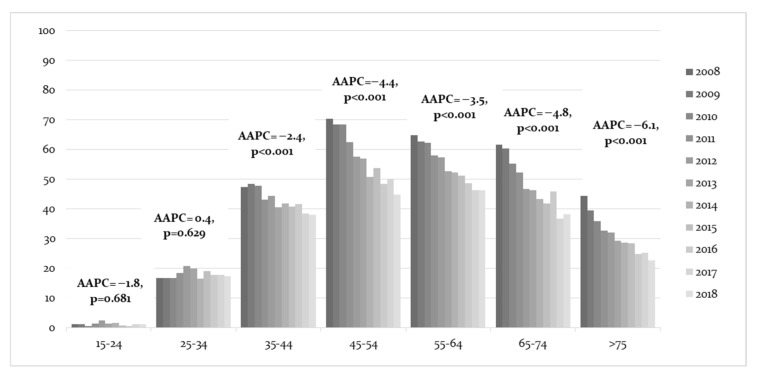
Average annual percent change (AAPC) of invasive cervical cancer trends in women’s classes of age and year regarding hospital admission, with related confidence intervals at 95%.

**Figure 3 cancers-15-01452-f003:**
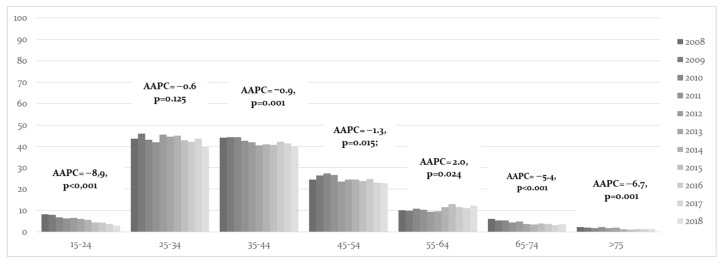
Average annual percent change (AAPC) of in situ cervical cancer trends understood via women’s classes of age and year of hospital admission, with related confidence intervals at 95%.

**Figure 4 cancers-15-01452-f004:**
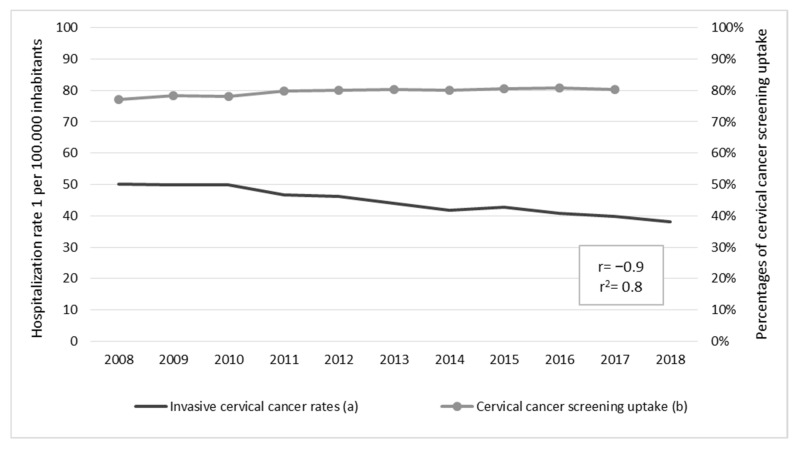
Correlation between hospitalization rates due to invasive cervical cancer and cervical cancer screening uptake with their respective average annual percent change (AAPC): (**a**) 2008–2018 AAPC = −2.8%, 95% CI = −3.2, −2.4, *p* < 0.001; (**b**) 2008–2018 AAPC = 0.5%, 95% CI = 0.2, 0.7, *p* < 0.001.

**Figure 5 cancers-15-01452-f005:**
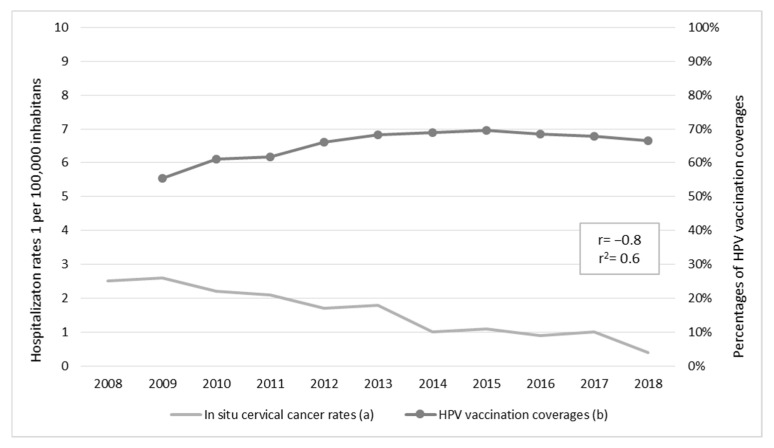
Correlation between hospitalization rates due to in situ cervical cancer and vaccination coverage against human papillomavirus (HPV), with their respective average annual percent change (AAPC): (**a**) 2008–2018 AAPC = −14.7%, 95% CI = −18.7, −10.5, *p* < 0.001; (**b**) 2009–2018 AAPC = 2.0%, 95% CI = 1.1, 2.9, *p* < 0.001.

**Table 1 cancers-15-01452-t001:** Socio-demographic and clinical characteristics, as reported in the Italian hospital discharge records (HDRs) due to all HPV-related diseases.

Characteristics	*n* (%)
Sex	
Female	446,435 (66.6)
Male	223,931 (33.4)
Age (median, IQR)	54 (41–68)
Marital status	
Single	133,713 (20.2)
Married	277,415 (42.0)
Separated	19,85 (3.0)
Divorced	16,682 (2.5)
Widow/widower	39,637 (6.0)
Not declared	173,924 (26.3)
Citizen	
Italian	617,826 (92.2)
Migrant residents in Italy	45,906 (6.9)
Refugees	6635 (1.0)
Length of hospital stay, in days (median, IQR)	6 (3–13)
HPV-related diseases	
Cervical cancer	291,368 (43.5)
Invasive cervical cancer	104,278 (15.6)
In situ cervical cancer	54,271 (8.1)
Anal cancer	42,127 (6.3)
Oropharyngeal cancer	234,652 (35.0)
Penile cancer	15,804 (2.4)
Vulvar cancer	34,510 (5.2)
Genital warts	59,449 (8.9)

**Table 2 cancers-15-01452-t002:** Hospitalization rates due to HPV-related diseases per 100,000 inhabitants in Italy 2008–2018.

Year	Female		Male	Both Sex		
	Cervical Cancer Rate (SD)	Vulva and Vagina Cancer Rate (SD)	Penile Cancer Rate (SD)	Anus Cancer Rate (SD)	Oropharynx Cancer Rate (SD)	Genital Warts Rate (SD)
2008	115.1 (0.9)	9.3 (0.3)	2.7 (0.1)	7.1 (0.2)	12.3 (0.3)	11.7 (0.3)
2009	109.7 (0.9)	9.0 (0.3)	2.7 (0.1)	6.7 (0.2)	11.8 (0.3)	11.6 (0.3)
2010	105.3 (0.9)	8.9 (0.2)	2.7 (0.1)	6.6 (0.2)	11.6 (0.3)	11.1 (0.3)
2011	96.2 (0.8)	9.3 (0.3)	2.8 (0.1)	6.3 (0.2)	10.8 (0.3)	10.3 (0.3)
2012	95.1 (0.8)	8.4 (0.2)	2.7 (0.1)	6.2 (0.2)	10.5 (0.3)	10.5 (0.3)
2013	90.5 (0.8)	8.0 (0.2)	2.8 (0.1)	5.7 (0.2)	9.9 (0.3)	10.0 (0.3)
2014	87.8 (0.8)	7.9 (0.2)	2.7 (0.1)	5.6 (0.2)	9.3 (0.3)	9.4 (0.3)
2015	85.8 (0.8)	8.4 (0.2)	2.7 (0.1)	5.8 (0.2)	9.1 (0.3)	8.9 (0.2)
2016	83.0 (0.8)	8.3 (0.2)	2.6 (0.1)	5.7 (0.2)	9.0 (0.3)	8.4 (0.2)
2017	81.3 (0.8)	8.2 (0.2)	2.6 (0.1)	5.9 (0.2)	8.3 (0.2)	8.5 (0.2)
2018	79.2 (0.7)	8.1 (0.2)	2.6 (0.1)	5.9 (0.2)	8.5 (0.2)	7.8 (0.2)
2008–2018	93.5 (0.8)	8.5 (0.2)	2.7 (0.1)	6.1 (0.2)	10.1 (0.3)	9.8 (0.3)

**Table 3 cancers-15-01452-t003:** Female patients’ median age and median regarding the length of hospital stay for cervical cancer during the years 2008–2018.

Year	Median Age (IQR)	*p*	Median Length of Stay (IQR)	*p*
2008	44 (35–55)	<0.001	3 (2–7)	<0.001
2009	44 (35–54)		3 (2–7)	
2010	44 (35–55)		4 (2–7)	
2011	44 (36–55)		4 (2–7)	
2012	44 (35–55)		4 (2–7)	
2013	44 (35–55)		4 (2–7)	
2014	44 (36–55)		4 (2–7)	
2015	45 (36–55)		4 (2–7)	
2016	45 (36–55)		4 (2–7)	
2017	45 (36–55)		4 (2–7)	
2018	46 (36–56)		4 (2–7)	

## Data Availability

Data will be available after sending a motivated request to corresponding author.

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
