# Peer review of "Impact of Preventive Strategies on HPV-Related Diseases: Ten-Year Data from the Italian Hospital Admission Registry"

_cancers, 2023, doi:10.3390/cancers15051452_

Round 1
Reviewer 1 Report
Restivo et al have examined public health data related to HPV disease and hospitalization trends in Italy from 2008-2018. They examine immunization coverage, adherence to cervical cancer screening and hospital admission for HPV related cancers.
Overall, this is an appropriate study and the basic design appears straight-forward. Significant reductions in cervical and head and neck cancers leading to hospitalization are reported over the study period. Correlations are observed between vaccine uptake and screening compliance.
Comments:
1) There are a lot of mistakes in the English usage. This is simply not written well enough to consider for publication. How are alcohol and tobacco voluptuous? "Dead female to cerivcal cancer" just reads oddly. What are cervical cancer screening adhesions? Do you mean adherence? Widower means male/Widow means female. Presumably they mean widow/widower in table 1? There are too many mistakes to comment on.
2) As stated in the discussion, a weakness of the work is that it is based on hospital admission and does not capture outpatient treatment. I would expect an increase in cervical cancer screening compliance to identify a lot of early stage disease that would require some clinical intervention, much the same way as adoption of regular mammograms resulted in a massive increase in the perceived incidence of breast cancer. Why is this not observed? I think its important they get some idea of outpatient treatment incidence. If that is going up, screening is working. If its not, the vaccine is working.
3) Is there some way to tell what fraction of oropharyngeal cancers in Italy are HPV+ vs HPV-? Is this available from the records they can access, either via HPV testing or P16 positivity by pathology staining as a surrogate marker? Other developed countries report values like 70%, which could lead to the data observed here. If it was only say 30%, this data could not possibly be real.
4) the average annualized percent changes reported for cervical cancer and oropharyngeal canner are very large. I have no way to determine if the statistics are calculated properly. Are other G7 nations seeing such profound effects? I am a little worried about the math here. It would be reassuring if similar changes are being observed elsewhere besides the Sicily study they reference.
5) The data shown in figures 4 and 5 simply does not look like a Pearson's r value of -0.8 or -0.9 to this statistically challenged reviewer. One curve is basically flat and the other is decreasing.
Author Response
Restivo et al have examined public health data related to HPV disease and hospitalization trends in Italy from 2008-2018. They examine immunization coverage, adherence to cervical cancer screening and hospital admission for HPV related cancers.
Overall, this is an appropriate study and the basic design appears straight-forward. Significant reductions in cervical and head and neck cancers leading to hospitalization are reported over the study period. Correlations are observed between vaccine uptake and screening compliance.
Comments:
1) There are a lot of mistakes in the English usage. This is simply not written well enough to consider for publication. How are alcohol and tobacco voluptuous? "Dead female to cerivcal cancer" just reads oddly. What are cervical cancer screening adhesions? Do you mean adherence? Widower means male/Widow means female. Presumably they mean widow/widower in table 1? There are too many mistakes to comment on.
- We really appreciated this observation about the use of English and we submitted the manuscript to a deeper grammar check by a native English language.
2) As stated in the discussion, a weakness of the work is that it is based on hospital admission and does not capture outpatient treatment. I would expect an increase in cervical cancer screening compliance to identify a lot of early stage disease that would require some clinical intervention, much the same way as adoption of regular mammograms resulted in a massive increase in the perceived incidence of breast cancer. Why is this not observed? I think its important they get some idea of outpatient treatment incidence. If that is going up, screening is working. If its not, the vaccine is working.
- We thank you so much for your smart remark about the HPV-relative diseases trend. We all agreed with you on the fact that outpatients’ clinical records helped us to have the real trend of HPV-related diseases in Italy, including also those requiring not intensive treatment. As Capra et al had already reported in their paper, precancerous cervical lesions such as ASC-US, LSIL and HSIL, are treated in outpatient healthcare settings, according to national laws. However no trend are reported about this source of data in Italy after preventive strategies implementation (Pubmed search strategy: HPV outpatient visit and Italy) and analyzing only the trend of outpatient setting could be not satisfactory for preventive measure effectiveness. Consequently, our focus was the hospitalizations for HPV-related diseases. Generally, more severe diseases need hospitalizations and their trend assessment let us understand the severity grade of HPV-related diseases. Indeed, the main objective of preventive strategies is reduce the burden of HPV related disease. If the trend decreases, this could mean the severity of HPV-related diseases is lowering, consequently preventive measures are working.
3) Is there some way to tell what fraction of oropharyngeal cancers in Italy are HPV+ vs HPV-? Is this available from the records they can access, either via HPV testing or P16 positivity by pathology staining as a surrogate marker? Other developed countries report values like 70%, which could lead to the data observed here. If it was only say 30%, this data could not possibly be real.
- Thank you for your appealing point of view. We collect the HDRs with diagnoses for HPV-related diseases. Unfortunately, HPV test results in HDRs were missing. Therefore, we did have enough data on oropharyngeal cancers with related HPV test results nor the related genotype. As far as the value of 30% is concerned, this is the reduction percentage of oropharyngeal cancer due HPV infection during the period 2008-2018, not the fraction of this cancer with HPV test positivity. To obtain the estimated rates for diseases highly probably caused by HPV infection, the attributable fractions were used. In detail for oropharyngeal cancers was used the review of Kreimer AR, Clifford GM, Boyle P et al. Human papillomavirus types in head and neck squamous cell carcinomas worldwide: a systematic review. Cancer Epidemiol Biomarkers Prev 2005;14(2): 467–75. doi:10.1158/1055-9965.EPI-04-0551 where it was reported an overall prevalence of 26%.
4) the average annualized percent changes reported for cervical cancer and oropharyngeal canner are very large. I have no way to determine if the statistics are calculated properly. Are other G7 nations seeing such profound effects? I am a little worried about the math here. It would be reassuring if similar changes are being observed elsewhere besides the Sicily study they reference.
- We already checked the results and we calculated the APC and AAPC of HPV-related diseases with the statistical software Joinpoint, in which hospitalization rates for each year were reported. The model selection method was Permutation Test, as Kim et al explain in their paper at reference no. 51. As far as hospitalization rates for HPV-related diseases in other G7 nations are concerned, we did not find any literature about in the same study period (2008-2018). Indeed, the period since the adoption of preventive strategies as vaccination and screening should be crucial in showing similar change in hospitalization rate. However it was reported in the discussion section the trend of HPV related hospital admission for Italy in the period 2001-2012 “Furthermore, the Italian reduction in HPV cervical cancer from 2008 to 2018 was slightly higher than that observed in the period 2001-2012 (APC= -2.9%, 95% CI= −3.8, −2.1)”.
5) The data shown in figures 4 and 5 simply does not look like a Pearson's r value of -0.8 or -0.9 to this statistically challenged reviewer. One curve is basically flat and the other is decreasing.
- Thank you so much for your comment. A factor that could help to explain the value obtained is the large number of observation obtained because it was reported 11 pairs of value that make more strong a value of correlation that graphically seems to be flat.
Reviewer 2 Report
The article by Vincenzo Restivo et al. conducted a retrospective study investigating HPV-related diseases based on national-wide hospital discharge records(HDR) from 2008 to 2018 in Italy. Over a decade, 670367 hospitalization records were found to be associated with HPV-related diseases. From the medical records, the authors revealed a significant decrease in hospitalization rates for multiple HPV-related cancers, such as cervical cancer, vulval and vaginal cancer et al. The authors summarized and concluded that HPV vaccination and routine cervical screening could help decrease HPV-related hospitalization rates. Overall this regional study provides evidence for the impact of preventive measures on hospitalization rates for cervical cancer.
1. Have you examed hospital discharge rates by conducting a geographical analysis? It could potentially visualize hospitalization patterns in different regions in Italy.
2. In the study design section, One of the Italy regions, Sicily, has been excluded from the analysis. Can you explain why this region is not included in this study?
3. The reduction of HDR in ten years may be due to an improvement in healthcare infrastructure. For example, higher qualified workforces and improved procedures for patients et al. Can you comment on this?
4. HPV vaccination and routine cervical screenings are standard preventive strategies to reduce cervical cancer risk worldwide. Therefore, this retrospective study does not provide novel insights into this disease. Can you also comment on this?
Author Response
The article by Vincenzo Restivo et al. conducted a retrospective study investigating HPV-related diseases based on national-wide hospital discharge records(HDR) from 2008 to 2018 in Italy. Over a decade, 670367 hospitalization records were found to be associated with HPV-related diseases. From the medical records, the authors revealed a significant decrease in hospitalization rates for multiple HPV-related cancers, such as cervical cancer, vulval and vaginal cancer et al. The authors summarized and concluded that HPV vaccination and routine cervical screening could help decrease HPV-related hospitalization rates. Overall this regional study provides evidence for the impact of preventive measures on hospitalization rates for cervical cancer.
- Have you examed hospital discharge rates by conducting a geographical analysis? It could potentially visualize hospitalization patterns in different regions in Italy.
- We performed the requested analysis by dividing Italian hospital admission for cervical cancer by the main three areas (Northern, Centre, and Southern Italy) as reported:
- Northern Italy:
- in situ cervical cancer: APC 2008-2018= -20,7% (IC 95%= -27,2%; -13,6%), p<0,001;
- invasive cervical cancer: APC 2008-2017= -4,4% (IC 95%= -5,2%; -3,6%), p<0,001;
- HPV vaccination coverage: APC 2009-2018= 1,6% (IC 95%= 1,0%; 2,2%), p<0,001;
- cervical cancer screening compliance: 1 Joinpoint: APC 2008-2015= 0,6% (IC 95%= 0,3%; 0,9%), p=0,002; APC 2015-2017= -5,5% (IC 95%= -7,5%; -3,5%), p=0,001;
- Centre Italy:
- in situ cervical cancer: 1 Joinpoint: APC 2008-2015= -5,5% (IC 95%= -23,0%; 16,0%), p=0,523; APC 2015-2018= -48,6% (IC 95%= -76,1%; 10,5%), p=0,077;
- HPV vaccination coverage: 1 Joinpoint: APC 2009-2013= 3,8% (IC 95%= 1,8%; 5,9%), p=0,004; APC 2013-2018= -1,1% (IC 95%= -2,4%; 0,3%), p=0,098;
- invasive cervical cancer: APC 2008-2017= -1,5% (IC 95%= -2,4%; -0,6%), p=0,005;
- cervical cancer screening compliance: APC 2009-2017= 0,5% (IC 95%= 0,2%; 0,7%), p=0,002;
- Southern Italy:
- in situ cervical cancer: APC 2008-2018= -5,8% (IC 95%= -10,1%; -1,4%), p=0,016;
- HPV vaccination coverage: APC 2009-2018= 1,2% (IC 95%= 0,0%; 2,4%), p<0,001;
- invasive cervical cancer: APC 2008-2017= -2,1% (IC 95%= -2,9%; -1,2%), p<0,001;
- cervical cancer screening compliance: 1 Joinpoint: APC 2009-2012= 1,4% (IC 95%= 0,7%; 2,0%), p=0,003; APC 2012-2018= 0,3% (IC 95%= -0,2%; 0,7%), p=0,208.
All the AAPC had a significant or marginal significant decrease that can us avoid any confounding effect for geographical residency.
- In the study design section, One of the Italy regions, Sicily, has been excluded from the analysis. Can you explain why this region is not included in this study?
- We thank you for this comment. The main reason for Sicily exclusion from the analysis is the previous publication of its data in 2019 (Restivo, V.; Costantino, C.; Amato, L.; Candiloro, S.; Casuccio, A.; Maranto, M.; Marrella, A.; Palmeri, S.; Pizzo, S.; Vitale, F.; et al. Evaluation of the Burden of HPV-Related Hospitalizations as a Useful Tool to Increase Awareness: 2007-2017 Data from the Sicilian Hospital Discharge Records. Vaccines (Basel) 2020, 8, E47, doi:10.3390/vaccines8010047), whose examined period was 2007-2017. To avoid the publication of duplicated data, we had decided to exclude Sicilian HDRs.
- The reduction of HDR in ten years may be due to an improvement in healthcare infrastructure. For example, higher qualified workforces and improved procedures for patients et al. Can you comment on this?
- We are right with you about the possible role in hospitalization change. As you had already written, the decrease of hospitalization rates due to HPV-related diseases could be even related to an improvement of healthcare services quality, including better procedures, well-planned workforces, and effective leadership. Further studies are needed to explore the weight of several factors, assessing all the aspects implicated into healthcare organization outcomes, also involving hospitalization rates for HPV-related diseases. However, no change in the appropriateness of HPV related diseases are occurred in the healthcare indications or general procedures during the study period in Italy. Even if the role of several factors in decreasing HPV related disease is not attributable due to the study design, the most probable is the role of preventive measures as screening and vaccination that were adopted more during the study period.
- HPV vaccination and routine cervical screenings are standard preventive strategies to reduce cervical cancer risk worldwide. Therefore, this retrospective study does not provide novel insights into this disease. Can you also comment on this?
- We appreciate your request that make us more clear the advantage of the study conducted. Indeed, there are two reasons to consider the novelty of data introduced with this manuscript. Firstly, the lack of data about the correlation between HPV-related diseases trend and preventive measures uptake, as well as the compliance with these, are still lower than the standard reference by National Plan for diseases prevention. Secondly, the cervical cancer has a long-term onset since HPV infection occurs and little data are published about long term effectiveness of preventive measures. Accordingly, the comparison between HPV vaccination/cervical cancer screening compliance and hospitalization rates due to cervical cancer let us evaluate the effectiveness of these preventive measures after many years since their adoption in Italy by reporting real world data.
Round 2
Reviewer 1 Report
The authors have made changes and included some of the missing information. I now understand that they adjust their patient code data by a factor to estimate the true numbers of HPV positive disease cases for the various sites based on their previous study. This is just an estimate and not necessarily an exact measure of HPV status. This is probably okay, but needs to be clearly stated as a limitation of the work.
Despite improvements in the writing, the manuscript is still not at a level I find acceptable for publication.
Author Response
Q: The authors have made changes and included some of the missing information. I now understand that they adjust their patient code data by a factor to estimate the true numbers of HPV positive disease cases for the various sites based on their previous study. This is just an estimate and not necessarily an exact measure of HPV status. This is probably okay, but needs to be clearly stated as a limitation of the work.
A: According to your comment we added a sentence in the limitation section as following “Moreover, the HPV attributable fraction of HDRs is not equivalent to an exact measure of HPV status.“
Q: Despite improvements in the writing, the manuscript is still not at a level I find acceptable for publication.
A: We already submitted the manuscript to a revision by a native English writer. Notwithstanding, it will have another revision by the editing service before manuscript publication.
Reviewer 2 Report
Thank you for the authors' detailed reply. I have no further concerns about the study.
Author Response
Q:Thank you for the authors' detailed reply. I have no further concerns about the study.
A:Thank for your comments that improved the quality of manuscript.